# Amelioration of LPS-Induced Jejunum Injury and Mucus Barrier Damage in Mice by IgY Embedded in W/O/W Emulsion

**DOI:** 10.3390/foods13244138

**Published:** 2024-12-20

**Authors:** Zhaohui Wang, Ruihua Ye, Shidi Zhang, Chuanming Liu, Ke Chen, Kongdi Zhu, Pengjie Wang, Fuqing Wang, Jiaqiang Huang

**Affiliations:** 1Key Laboratory of Precision Nutrition and Food Quality, Department of Nutrition and Health, Ministry of Education, China Agricultural University, Beijing 100083, China; wzhaohui0222@163.com (Z.W.); sy20233313708@cau.edu.cn (S.Z.); liucm0501@163.com (C.L.); 15529082968@126.com (K.C.); zkd15938702692@163.com (K.Z.); wpj1019@cau.edu.cn (P.W.); 2College of Veterinary Medicine, China Agricultural University, Beijing 100083, China; yerh0522@cau.edu.cn; 3Tibet Tianhong Science and Technology Co., Ltd., Lasha 851414, China

**Keywords:** IgY, LPS, jejunum injury, mucous membrane barrier, bioavailability, immunotherapy, intestinal health

## Abstract

Chicken yolk immunoglobulin (IgY) is a natural immunologically active antibody extracted from egg yolk and can be used as a natural dietary supplement for the treatment of inflammation and damage to the intestines. In our study, IgY was embedded in a double emulsion (W/O/W; DE) to explore the therapeutic effect of the embedded IgY on Lipopolysaccharide (LPS)-induced jejunal injury in mice. The results showed that W/O/W-embedded IgY as a dietary supplement (IgY + DE) attenuated LPS-induced damage to mouse small intestinal structures and protected the integrity of the jejunal mucosal barrier. IgY + DE increased the amount of related transcription factors (Math1, Spdef, Elf3, and Klf4) and promoted thrush cell differentiation. IgY + DE ameliorated LPS-induced reduction in mucin quantity and markers. It promoted the expression of Muc1 and Muc2 and increased the mRNA expression levels of Muc1, Muc2, Muc3, Muc4, Muc13, and Agr2 (*p* < 0.05). IgY + DE increased the expression of several glycosyltransferases involved in mucin glycosylation. IgY + DE also neutralized the LPS attack on the expression of jejunal inflammatory factors IL-1β, IL-6, IL-4, and TNF-α. In conclusion, the IgY-embedded double emulsion can be used as a dietary supplement for immunotherapy to prevent LPS-induced jejunal injury in mice.

## 1. Introduction

Lipopolysaccharide (LPS) is a complex molecule found mainly in the cell walls of Gram-negative bacteria and has a strong immunostimulatory effect [1]. LPS damages intestinal epithelial cells, resulting in impaired intestinal barrier function and increased intestinal permeability. LPS induces a complex series of inflammatory responses through activation of macrophages and other immune cells in the nuclear factor-κB (NF-κB) pathway [2], destroying the structure of the intestine and the intestinal barrier [3,4]. Chicken yolk immunoglobulin (IgY) is an immunoreactive antibody secreted by the autoimmune system in poultry egg yolks [5], which can be used for the protection and treatment of intestinal diseases in humans and animals. IgY consists of two heavy chains (H), two light chains (L), and four constant regions (CH1-CH4), with a Y-shaped molecular structure [6]. Currently, IgY is widely used in foods and nutraceuticals to enhance human immunity and resistance [7]. IgY can be used to treat gastrointestinal bacterial infections, such as those caused by *Salmonella* and *Helicobacter pylori* [6,8]. A number of studies have shown that IgY can reduce endotoxin damage and ameliorate the inflammatory response by neutralizing bacterial endotoxins in the body [8]. The advantages of IgY in immunotherapy compared to traditional drugs for enteritis include the following: the safety of IgY, an immunological antibody in egg yolk from natural and healthy sources with good antigen binding and local activity [9]; the specificity of IgY, which acts on antigenic epitopes of specific pathogens, does not lead to the development of immune resistance, and does not affect the function of beneficial flora in the host [10]; and Fc fragments of IgY do not bind to rheumatoid factor or IgG, and do not activate human complement [11], so that IgY by itself does not lead to the development of immune resistance in the organism [12].

Therefore, IgY is a beneficial food for regulating intestinal health to intervene in LPS-induced enteritis. However, it is difficult for IgY to survive in low-acid environments and it is destroyed by acidic gastric juices and pepsin when passing through the gastric environment [6]. In addition, biological barriers [13] also lead to a decrease in the biological activity of the IgY antibody, which is one of the current technical challenges in IgY research and application. The existing technology is to use liposomes [14] and chitosan–sodium alginate drug-carrying nanoparticles [15] to encapsulate IgY, but liposome encapsulation cannot be applied to large-scale industrial food production, and the process of chitosan–sodium alginate drug-carrying nanoparticles is complicated and costly. In our previous study, we used W/O/W encapsulation of IgY to achieve stable delivery of IgY to the intestine [16]. In this study, IgY was embedded using a W/O/W emulsion to avoid degradation and inactivation of IgY in the gastric environment, in order to achieve the targeted delivery of food-borne IgY to the intestine.

In conclusion, we used W/O/W-embedded IgY technology to deliver IgY stably to the jejunum (1) to determine the ameliorative effect of IgY encapsulated in a W/O/W system on structural damage in the small intestine; (2) to study the effect of IgY encapsulated in the W/O/W system on goblet cells and mucosal proteins, and its protective effect against jejunal mucosal injury; and (3) to investigate the protective effect on inflammatory response in the jejunum.

## 2. Materials and Methods

### 2.1. Construction of Embedded IgY Double Emulsion

The construction of W/O/W emulsions embedded with IgY and their stability analysis were performed as described previously [16]. The emulsifier PCPR (4% *w*/*w*) was added to corn oil to form the oil phase (McLean, Shanghai, China), and an amount of 15 g [17] IgY (Unik, Beijing, China), NaCl (0.2% *w*/*w*), and 5 g sorbitol (Biotopped, Beijing, China) were added to the internal aqueous phase. The water and oil phases were mixed in a 2:3 ratio and sheared at 11,000 rpm for 3 min using a high-shear mixer (T25 digital ULTRA-TURRAX, IKA, Shanghai, China) to form the primary emulsion. The primary emulsions were then homogenized at 60 MPa (NS 1001, GEA Niro Soavi, Parma, Italy) to form W/O emulsions.

Tween-80 (T80) (4% *w*/*w*) (Macklin, Shanghai, China), lecithin (PC) (4% *w*/*w*) (BZ52097, Biotopped, Beijing, China), sodium caseinate (SC) (4% *w*/*w*) (Biotopped, Beijing, China), and whey protein powder (WPI) (4% *w*/*w*) (Biotopped, Beijing, China) were added to water to form four different external aqueous phases. The final W/O/W emulsion was then made by mixing the initial W/O emulsion with the external water phase in a 2:3 ratio using a high-shear mixer (T25 Digital ULTRA-TURRAX, IKA, Shanghai, China) and shearing at 10,000 rpm for 3 min.

### 2.2. Animal Ethics Statement

The experimental protocol was approved by the Ethics Committee of Animal Experimentation of China Agricultural University, with the approval date of 23 May 2024, and the approval number AW32504202-5-5 (Beijing, China).

KM mice have a wide range of applications in drug screening and disease modeling. The 50 male KM mice used in the experiment were purchased from Spectrum, Beijing, China. All animals were housed under SPF conditions with 12 h of light and 12 h of darkness, and fed and watered ad libitum at 22 °C. Mice were acclimatized for 7 days before the start of the experiment.

The 50 KM mice were divided into 5 groups. These were as follows: control group (Control; *n* = 10); LPS group (LPS; *n* = 10); IgY group (IgY; *n* = 10); DE group (DE; *n* = 10); and IgY + DE group (IgY + DE; *n* = 10). As shown in Figure 1, the mice were administered by gavage for 14 consecutive days. Mice were injected intraperitoneally with LPS on day 15. Gavage administration and intraperitoneal injection doses are shown in Table 1. The dose of IgY was 48 mg/mL. Mice were gavaged using a straight-tip 10-gauge gavage needle (GWZ110). Samples were delivered directly into the mouse stomach in contact with gastric hydrochloric acid.

At the end of the LPS injection, 50 mice were subjected to cervical dislocation and then analyzed. Small bowel sections of 3–5 mm were rinsed with PBS and then fixed with paraformaldehyde for subsequent histological analysis; the remaining small intestine tissue was stored frozen at −80 °C for subsequent analysis.

### 2.3. H&E and AB-PAS Staining

Small intestinal tissues fixed in 4% paraformaldehyde were embedded in paraffin for hematoxylin–eosin (H&E) staining. Villus height (VH), crypt depth (CD), and the VH/CD ratio were measured using Image J (version 1.48r, National Institutes of Health, Bethesda, MD, USA). Alcian Blue Periodic Acid–Schiff (AB-PAS) staining stains goblet cells in sectioned tissue blue. The AB-PAS staining technique is a histological technique that combines two staining methods for the detection and visualization of glycans, especially acidic mucins and neutral glycans, in tissue sections. This staining method is very useful in pathologic studies and clinical diagnosis, especially in the assessment of mucus secretion and glycan distribution in tissues. Blinded analysis was performed using Image J (version 1.48r, National Institutes of Health, Bethesda, MD, USA) software. The proportion of goblet cells was quantified based on the ratio of blue area to total area.

### 2.4. Immunohistochemistry

Sections were deparaffinized, rehydrated, and then immersed in sodium citrate buffer to expose heat-induced epitopes. Sections were incubated with 10% Goa serum for 1 h. The epitopes were then incubated with Muc1 antibody (1:800; GB114938, Servicebio Technology Co., Ltd. Wuhan, China) and Muc2 antibody (1:1000; GB11344, Servicebio Technology Co., Ltd. Wuhan, China) overnight. Tissue sections were incubated overnight with biotinylated goat anti-rabbit IgG secondary antibody (1:500, A0277, Beo Timeless Co., Ltd., Beijing, China). The tissues were then rinsed with PBS and incubated with 1:300 HRP–Streptavidin (1:400, A0303, Beo Timeless Co. Ltd., Beijing, China) for 45 min.

### 2.5. RNA Isolation and RT-qPCR Analysis

Total RNA was extracted by mixing TRIzol reagent with jejunal tissue (CW0580; Coyne Biotechnology, Beijing, China). RNA was reverse-transcribed to generate cDNA using a reverse transcription kit (gDNA digester plus) (11141ES10; Yeasen Biotechnology Co., Ltd., Shanghai, China). qPCR amplification was performed using the Step One Plus Real-Time PCR System (Applied Biosystems, Waltham, MA, USA). Primer sequences for RT-qPCR are listed in Table 2.

### 2.6. ELISA

An amount of 100 mg of jejunal tissue and PBS were used for tissue homogenization. The homogenate was ground at 65 Hz for 5 min. After homogenization, the tissue was centrifuged at 5000 g rpm for 10 min and then the supernatant was removed. The assay was performed using an Elisa kit for quantitative analysis of IL-1b, TNF-a, IL-6, IL-10, and IL-4 (Jiangsu Jingmei Biotechnology Co., Ltd., Yanchen, China).

### 2.7. Statistical Analysis

All parameters are expressed as mean ± SD. *p* < 0.05 was the criterion for significance. Analyses were performed using the GraphPad Prism 10.0 program (GraphPad Software Co., Inc., San Diego, CA, USA). Multiple group comparisons were performed using Tukey’s multiple comparison test and one-way ANOVA.

## 3. Results

### 3.1. IgY + DE Alleviates LPS-Induced Structural Damage of Small Intestinal Tissues

According to the results, the duodenum, jejunum, and ileum tissues showed different degrees of damage and destruction after LPS stimulation compared to the LPS group. This was attenuated by the IgY + DE intervention (Figure 2a). In the reference Control group, the length of villi in the duodenum (*p* < 0.0001), ileum (*p* < 0.001), and jejunum (*p* < 0.0001) was significantly shorter after LPS induction (Figure 2b–d). The same trend was observed in the IgY and DE groups. This phenomenon of shortening of villi length was significantly alleviated after IgY + DE intervention in the duodenum (*p* < 0.01), ileum (*p* < 0.05), and jejunum (*p* < 0.0001) compared to the LPS group (Figure 2b–d). Meanwhile, the depth of the crypts in the duodenum (*p* < 0.05), jejunum (*p* < 0.05), and ileum (*p* < 0.001) were significantly increased in the LPS group compared to the Control group (Figure 2e–g), resulting in a significant the decrease in the VH/CD. The IgY + DE intervention attenuated the increase in the duodenum, ileum (*p* < 0.001), and jejunum (*p* < 0.05) crypt depth and decrease in the V/C ratio associated with LPS stimulation (Figure 2h–j). The results confirmed that LPS stimulation leads to varying degrees of injury in small bowel segments, and IgY + DE intervention could effectively alleviate the damage caused by LPS induction. Because LPS stimulation causes the most severe damage to the jejunum, we focused on the effects on the jejunum in the next experiments.

### 3.2. IgY + DE Increased Associated Transcription Factors and Goblet Cell Differentiation

To investigate LPS induction and IgY + DE expression of several transcription factors associated with the intervention of goblet cells in the production and secretion of the jejunum mucus layer, we measured the relevant mRNAs. The results indicated that IgY + DE intervention altered the mRNA expression of some relevant transcription factors involved in jejunum cell differentiation. The results established an increase in the expression of secretory cell markers in the jejunum, i.e., Math1 (*p* < 0.0001) and Spdef (*p* < 0.001) (Figure 3c,e). Markers involved in the terminal differentiation of secretory cells to follicular cells (i.e., Elf3 (*p* < 0.01) and Klf4 (*p* < 0.001)) were also significantly increased (Figure 3f,g). Decreased Hes1 mRNA expression was observed (Figure 3d). Due to the increased mRNA expression of these goblet cell transcription factors, we hypothesized that the number of goblet cells might also increase. To verify this, we measured the number of goblet cells (blue area) and found that the number of goblet cells was significantly increased in the IgY + DE-supplemented group compared to LPS (Figure 3a,b).

### 3.3. IgY + DE Ameliorated the LPS-Induced Reduction in Mucin Number and Markers

Next, we explored whether the increase in the number of goblet cells induced by IgY + DE was associated with an increase in the production of secreted mucin, resulting in the formation of a protective intestinal mucus layer. Muc2 and Agr2 are essential components required for the production and secretion of Muc2. We found that LPS stimulation resulted in decreased expression of Muc2 (*p* < 0.001) and Agr2 (*p* < 0.05), and supplementation with IgY + DE significantly increased the expression of Muc2 (*p* < 0.01) and Agr2 in the mucus layer (Figure 4e,f). Staining with Muc1 and Muc2 antibodies in the mucosal region showed that mucins Muc1 (*p* < 0.0001) and Muc2 (*p* < 0.0001) were significantly reduced in the LPS group, and their expression increased (*p* < 0.0001) after IgY + DE intervention (Figure 4a–d). We measured the mRNA expression of other mucins in the jejunal mucus layer. Muc1, Muc3, Muc4, and Muc13 are transmembrane mucins that play a protective and signaling role. The results confirmed that LPS induction reduced the expression of Muc1 (*p* < 0.001), Muc3 (*p* < 0.01), Muc4 (*p* < 0.01), and Muc13 (*p* < 0.001), and supplementation with IgY + DE significantly increased Muc1 (*p* < 0.001), Muc3 (*p* < 0.05), Muc4 (*p* < 0.01), and Muc13 (*p* < 0.05) expression in the jejunum (Figure 4g–j).

### 3.4. IgY + DE Increased the Expression of Glycosyltransferases Involved in Mucin Glycosylation

IgY + DE intervention promoted the expression of a number of related glycosyltransferases. In addition to markers associated with goblet cells and mucin production, we also studied the expression of a number of glycosyltransferases. Mucins are important components of the intestinal mucus layer. Glycosyltransferases can influence the composition of the mucus layer by constructing multiple glycans into the core components that make up mucin [18,19]. By measuring the expression of glycosyltransferases involved in the process of elongation and branching, the results confirmed a significant increase in Gcnt1 (*p* < 0.0001) and Gcnt4 (*p* < 0.01) in the jejunum of the IgY + DE group compared with that of the LPS-induced group (Figure 5a,b). Galactosyltransferase C1galt1 (*p* < 0.01) and its synthase C1galt1c1 were significantly increased in the jejunum (Figure 5c,d). Treatment with IgY + DE also affected the expression of glycosyltransferases involved in sugar chain termination. Indeed, the results show a significant increase in the expression of the fucoidan glycosyltransferases Futl1 and Fut2 in the jejunum (*p* < 0.05) (Figure 5e,f). Finally, the results established a significant decrease in the expression of St6galnac2 in the jejunum (Figure 5k).

### 3.5. IgY + DE Inhibits LPS-Induced Expression of Inflammatory Factors in Mouse Jejunum

First, we examined the mRNA expression of inflammatory factors in the jejunum. It was found that mRNA expression of IL-1b (*p* < 0.0001), IL-6, TNF-a (*p* < 0.0001), and IL-17a (*p* < 0.0001) was significantly increased in the LPS group compared to the control group, while mRNA expression of IL-10 (*p* < 0.05) was decreased. In contrast, mRNA expression of IL-6 (*p* < 0.01), IL-1b (*p* < 0.0001), TNF-a (*p* < 0.0001), and IL-17a (*p* < 0.0001) was decreased, and mRNA expression of IL-10 was increased after IgY + DE intervention (Figure 6a–e). To further assess the effect of IgY + DE intervention on LPS-induced jejunal inflammation, we examined the expression of inflammatory factors IL-1β, IL-10, IL-6, IL-4, and TNF-α. After IgY + DE intervention, the expression of IL-1β (*p* < 0.0001), IL-10 (*p* < 0.05), and IL-6 (*p* < 0.05) was suppressed, and the expression of IL-4 (*p* < 0.0001) and TNF-α (*p* < 0.0001) was promoted (Figure 6f–j).

## 4. Discussion

In this study, we found that treatment with IgY embedded in W/O/W counteracted the inflammatory response in the jejunum induced by LPS stimulation and found that these effects were associated with structural damage to the intestine; variations in the intestinal mucus layer barrier and mucin glycosylation are associated.

Intestinal morphology is one of the behavioral markers for the evaluation of inflammation [20]. Previous studies have demonstrated that LPS attack causes damage to intestinal structures, as evidenced by increased CD, decreased VH, and a decreased VH/CD ratio [21]. Similarly, the same trend was found in our study. CD, small intestinal VH, and the ratio of VH/CD are important indicators of the digestive and absorptive function of the small intestine [22]. This was significantly improved after IgY + DE intervention, which decreased small intestinal CD and increased VH and the VH/CD ratio. In addition, it has been shown that improved intestinal morphostructure, including longer intestinal VH, CD, and a higher ratio of VH/CD, contributes to stress relief and improved intestinal barrier function [23].

Gut health is critical and requires multiple layers of defense to deal with challenges. The tight junctions of epithelial cells form the first physical barrier, and the intestinal mucus layer acts as a second barrier to further maintain gut health. Distributed between monolayers of columnar cells, goblet cells act as specialized intestinal epithelial cells that secrete mucus and defend against bacterial invasion [24]. In addition, goblet cells are necessary for maintaining the integrity of the intestinal epithelium and mucosal barrier [25,26]. By first studying goblet cells, the results of the experiment showed that IgY + DE not only significantly increased the expression of transcription factors involved in goblet cell differentiation, but also increased the number of goblet cells in the jejunum. According to the results, markers indicating the differentiation of intestinal progenitor cells to secretory cells were perceived to be significantly reduced in the jejunum, Math1 (*p* < 0.0001) and Hes1 (*p* < 0.001), whereas both markers were significantly increased (*p* < 0.0001) after IgY + DE intervention. Inhibition of the Notch pathway in intestinal epithelial cells activates Math1, whereas activation of the Notch signaling pathway activates Hes1 [27]. Activation of the transcription factors Elf3 and Klf4 caused terminal differentiation of goblet cells. The markers Klf4 (*p* < 0.001) and Elf3 (*p* < 0.01) are involved in the terminal differentiation of secretory cells to alveolar cells and were increased in the jejunum after IgY + DE intervention. Improved differentiation of goblet cells not only helps to enhance intestinal barrier function and reduce inflammation, but also promotes micro-ecological balance and nutrient absorption, enhances the immune response, and helps to prevent a variety of intestinal-related diseases.

Mucins covering the surface of the intestinal epithelium are an important component of mucus layer formation, and the mucus barrier plays a key role in the chemical barrier of the intestine. The amount and maturity of mucin affect the ability of intestinal tissues to withstand risks [28]. We next explored whether the increase in the number of goblet cells induced by IgY + DE is associated with increased secretory mucin secretion, which forms a protective intestinal mucus layer. Comparing the LPS and control groups, the relative mRNA expression of Muc1 and Muc2 was significantly reduced after LPS induction. It has been shown that LPS stimulation can lead to mucin overproduction in the intestinal mucosal barrier, thus causing bacterial inflammation of the mucosa [29]. Supplementation with IgY + DE is sufficient to significantly increase the expression of the major mucus layer components, pre-secretory gradient 2 (Agr2), and mucin 2 (Muc2), which is transcriptionally involved in Muc2 synthesis and secretion. The protein disulfide isomerase Agr2 is essential for the production of intestinal mucus. When jejunum staining of mucosal areas was performed with Muc2 antibody, Muc2 expression was significantly higher in the IgY + DE group versus the LPS group. Transmembrane mucins are also important components of the mucus layer, where they protect the intestinal surface and carry out intracellular signaling [30]. Therefore, we measured the mRNA expression of some of them and observed that supplementation with IgY + DE significantly increased Muc1 expression in the jejunum. Furthermore, when jejunum staining of mucosal areas was performed with Muc1 antibody, Muc1 expression was significantly higher in the IgY + DE group versus the LPS group. Thus, IgY + DE contributes to the maintenance of jejunum epithelial barrier function in mice, which may be mainly related to the reduction in jejunum inflammation as well as changes in jejunum mucosa-associated microorganisms and their metabolites. Improving mucin production is important for enhancing intestinal barrier function, reducing inflammation, enhancing immune response, and disease prevention. Therefore, research and development of strategies to improve mucin production have important clinical and nutritional applications.

Next, we looked at some of the markers involved in mucin glycosylation that are altered by dietary factors [31,32,33,34,35]. Some reports suggest that mucin changes with dietary structure and nutrient composition [19]. One of the important major components of mucin is Muc2 [36]. Muc2 is synthesized and secreted by goblet cells and is rich in tandem repeat sequences of proline, threonine, and serine (PTS), referred to as the PTS structural domain. In particular, we focused on the glycosyltransferases associated with Muc2 glycosylation [19]. Surprisingly, we found that IgY intervention significantly regulated many glycosyltransferases in the jejunum. However, the correlation between IgY and mucin glycosylation has been less studied. According to the experimental results, the galactosyltransferase C1galt1 (*p* < 0.01) was significantly increased in the jejunum after IgY + DE intervention. A study has observed that mice with C1galt1-deficient intestinal epithelial cells inducing colitis grow up to develop colon tumors [37]. The high embryonic lethality of C1galt1 knockout mice prevented the development of a valid animal model of C1galt1 deficiency. At the same time, this suggests that C1galt1 and O-glycosylation are essential for normal development. It has also been reported that C1galt1 plays an important role in the development and progression of colorectal cancer (CRC) by participating in various molecular mechanisms [37]. In addition, the findings showed a significant increase in the expression of fucosyltransferase Fut2 in the jejunum (*p* < 0.05). The Fut2 gene has been reported to play a key role in the response against infection-related injury, gastric acid secretion, and gastrointestinal motility [38]. Indeed, much evidence suggests that the expression of some glycosyltransferases is altered in the development and progression of colorectal cancer and inflammation. Our study suggests that the dietary addition of IgY promotes the expression of some glycosyltransferases, which in turn promotes intestinal mucin secretion, relieves intestinal inflammation, and enhances intestinal barrier function.

Second, we further explored IgY + DE and LPS-induced damage to the intestinal mucosal barrier. Inflammatory factors have been identified as important contributors to intestinal epithelial barrier dysfunction and increased permeability [39]. LPS triggers immune system activity. Next, innate immune cells are activated to produce large amounts of pro-inflammatory and anti-inflammatory cytokines [4]. TNF-α, IL-6, and IL-1β are macrophage-derived cellular messengers that play important roles in a range of inflammatory responses [40]. IL 17 and IL 23 signaling have been reported to result in a cascade of TNF, IL-1b, and LPS pro-inflammatory molecules. It is well known that IL 17 and TNF synergistically mediate signals that drive inflammatory gene expression [41]. According to the results, LPS stimulation promoted the mRNA expression level of IL-17a. This may have caused a cascade response of LPS with pro-inflammatory factors. TNF-α impairs the integrity of the entero-mechanical barrier by inhibiting the expression of occludin and claudin-1 in the intestine [42]. In addition, elevated levels of IL-1β have been associated with disruption of Caco2 cell barrier function, leading to increased intestinal permeability [43]. In our study, we found increased expression of TNF-α, IL-1β, and IL-6 in the jejunum of LPS-stimulated mice. Consistent with our results, similar results were observed in a previous study in which mice had elevated concentrations of the inflammatory factors TNF-α, IL-1β, and IL-6 upon LPS attack [44]. Studies have shown that LPS-stimulated inflammatory injury is mediated through the TLR4/NFκB/NLRP3 signaling pathway [45]. And there is a close relationship between intestinal inflammation and the intestinal barrier. IL-6 and IL-1β have been reported to cause the rearrangement of tight junction proteins to impair intestinal barrier function, and TNF-α affects the tight junctions between intestinal epithelial cells [46]. After the IgY intervention, the inflammatory response was significantly improved. Consistent with previous results [47]. Thus, IgY + DE can attenuate LPS stimulation-induced inflammation in the mouse jejunum and help maintain jejunum mucosal barrier function.

The strong immunostimulatory effects of LPS can trigger a range of intestinal disorders. The main drugs available for the treatment of LPS-related diseases include antibiotics and nonsteroidal anti-inflammatory drugs (NSAIDs) [48]. Antibiotics can be used to control the release of LPS due to bacterial infections. However, antibiotics may lead to resistance, alteration of the intestinal flora, and secondary infections. Nonsteroidal anti-inflammatory drugs such as aspirin and ibuprofen may reduce inflammatory symptoms caused by LPS. However, long-term use may cause side effects such as gastrointestinal bleeding and kidney impairment. In addition to the use of antibiotics and NSAIDs for the treatment of enteritis, some trace elements such as selenium and its protein SELENOI have been shown to be effective in reducing intestinal inflammation [49,50,51,52]. Compared with existing treatments for LPS, IgY has advantages due to its natural origin, high immunological activity, and lack of toxic side effects [53]. These advantages have led to the widespread use of IgY as a novel immunotherapy in many disease areas.

## 5. Conclusions

These results indicate that, as a dietary supplement for regulating intestinal health, W/O/W-embedded IgY can improve the bioavailability of IgY to facilitate the supplementation of immune nutrients in the body. IgY, when released in the intestine, is protective against LPS-induced structural damage in the small intestine and jejunal barrier damage in mice. This provides a basis for further understanding of the beneficial effects of IgY and a new idea for the treatment of intestinal inflammatory response and mucosal injury. W/O/W-embedded IgY offers new perspectives on the development of non-invasive and more accessible treatments for enteritis, whose clinical applications need to be investigated in more advanced models or in humans. In addition, the simple preparation process of W/O/W-embedded IgY and the low cost of raw materials allow for large-scale industrial food production as dietary supplements or functional foods.

## Figures and Tables

**Figure 1 foods-13-04138-f001:**
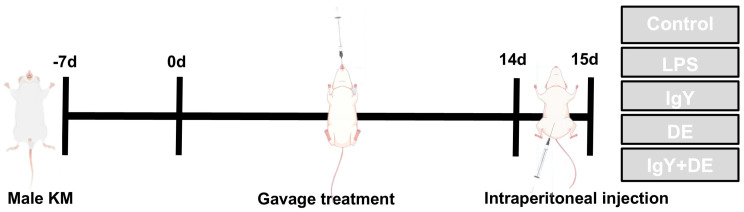
Experimental research design. After a 7-day acclimatization period, the mice were treated by gavage. Gavage was performed daily from 0 to 14 d. The mice were treated by gavage at the end of the 7-day acclimatization period. At the end of the gavage, mice were injected intraperitoneally at 15 d. The mice were then treated with a daily gavage treatment from 0 to 14 d.

**Figure 2 foods-13-04138-f002:**
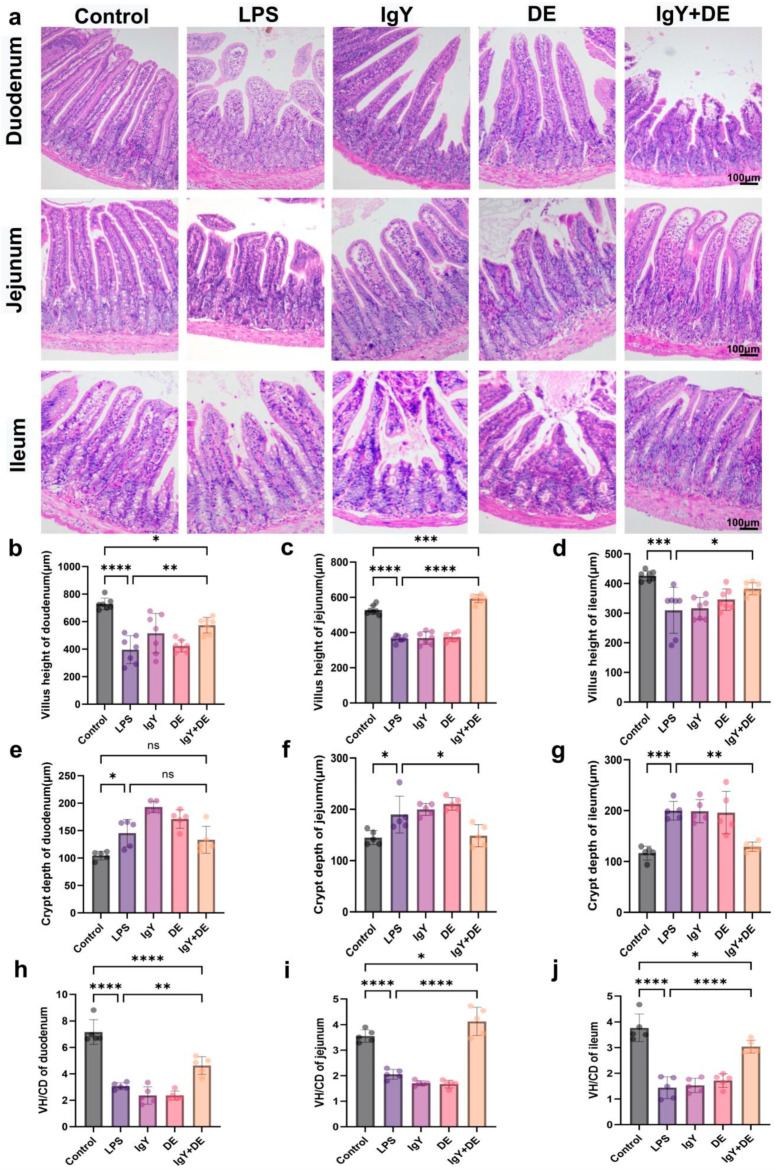
IgY + DE alleviates LPS-induced structural damage of small intestinal tissues. (**a**) H&E staining of duodenum, jejunum, and ileum (n = 5) (scale bar: 100 μm). (**b**) Villus heights of duodenum (n = 7). (**c**) Villus heights of jejunum (n = 7). (**d**) Villus heights of ileum (n = 7). (**e**) Crypt depth of duodenum (n = 7). (**f**) Crypt depth of jejunum (n = 7). (**g**) Crypt depth of ileum (n = 7). (**h**) V/C ratio in duodenum (n = 7). (**i**) V/C ratio in jejunum (n = 7). (**j**) V/C ratio in ileum (n = 7). * *p* < 0.05; ** *p* < 0.01; *** *p* < 0.001; **** *p* < 0.0001; ns: represents no significance between the values. Control: 0.9% NaCl gavage; LPS: 0.9% NaCl gavage; IgY: unembedded IgY gavage; DE: W/O/W gavage; IgY + DE: W/O/W gavage with embedded IgY.

**Figure 3 foods-13-04138-f003:**
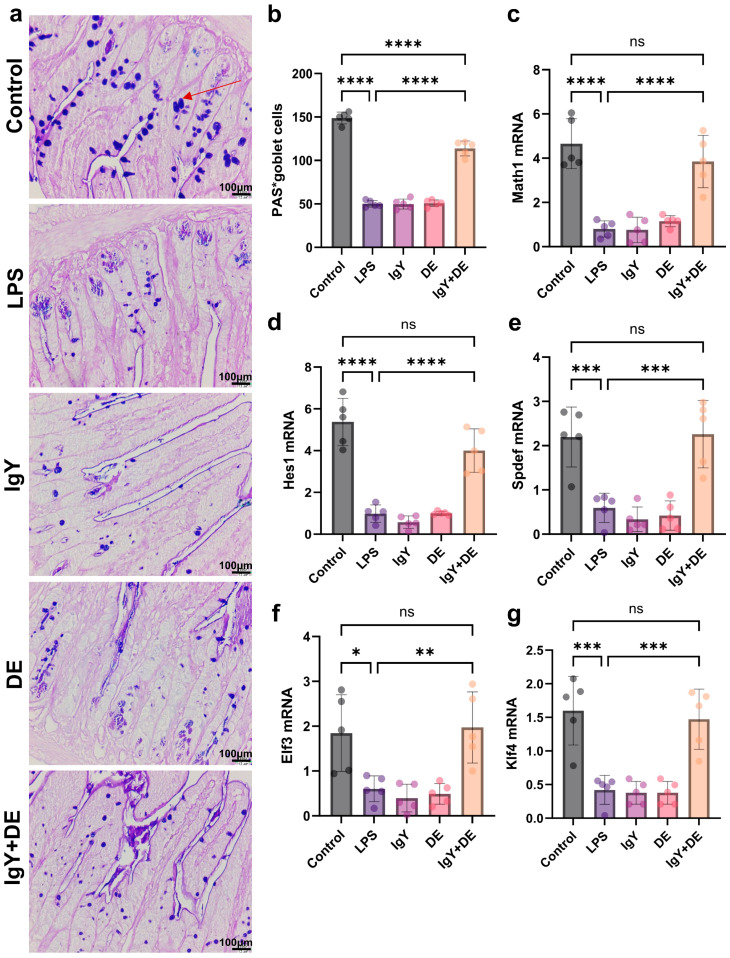
IgY + DE increased associated transcription factors and goblet cell differentiation. (**a**) AB-PAS staining of the jejunum (scale: 100 µm) (n = 5) (red arrows sign goblet cells.). (**b**) The number of goblet cells expressed as positive cells per villus in jejunum. (**c**–**g**) Relative mRNA expression of transcription factors involved in goblet cell differentiation in the jejunum. (**c**) Math1 mRNA expression (n = 5). (**d**) Hes1 mRNA expression (n = 5). (**e**) Spdef mRNA expression (n = 5). (**f**) Elf3 mRNA expression (n = 5). (**g**) Klf4 mRNA expression (n = 5). Atonal bHLH transcription factor 1 (Math1), his family bHLH transcription factor 1 (Hes1), SAM pointed domain containing ETS transcription factor (Spdef), E74-like ETS transcription factor 3 (Elf3), kruppel-like factor 4 (Klf4). * *p* < 0.05; ** *p* < 0.01; *** *p* < 0.001; **** *p* < 0.0001; ns: represents no significance between the values. Control: 0.9% NaCl gavage; LPS: 0.9% NaCl gavage; IgY: unembedded IgY gavage; DE: W/O/W gavage; IgY + DE: W/O/W gavage with embedded IgY.

**Figure 4 foods-13-04138-f004:**
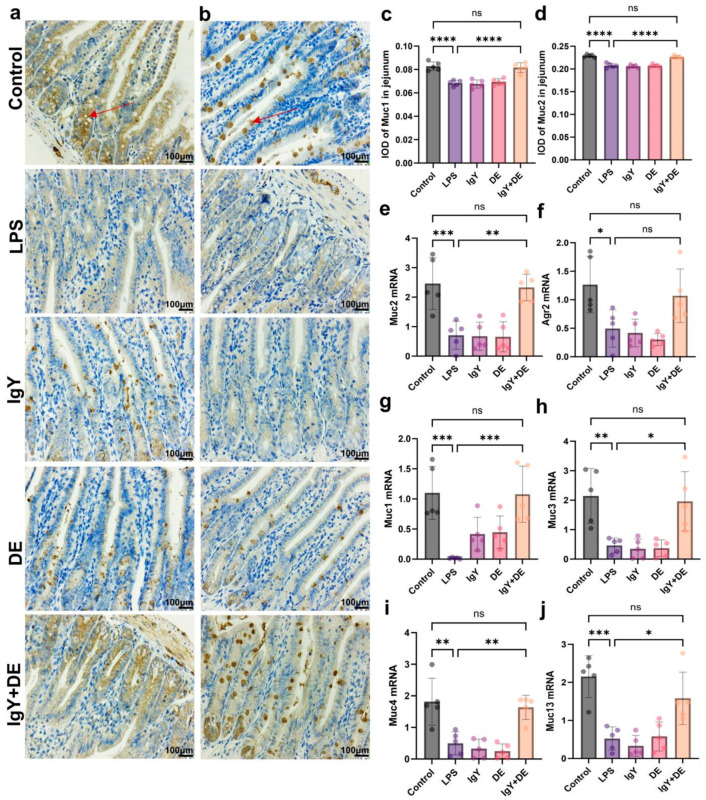
IgY + DE ameliorated the LPS-induced reduction in mucin number and markers. (**a**) Immunohistochemical staining of Muc1 in jejunum sections (scale bar 100 µm) (n = 5) (red arrows indicate Muc1). (**b**) Immunohistochemical staining of Muc2 in jejunum sections (scale bar 100 µm) (n = 5) (red arrows indicate Muc2). (**c**) IOD values of jejunum Muc1-positive cells. (**d**) IOD values of jejunum Muc2-positive cells. The IOD value refers to the integral optical density, which is a numerical indicator of the brightness or intensity of a specific stained area in a tissue section quantified by image analysis techniques. (**e**–**j**) Relative mRNA expression of markers involved in mucin production in the jejunum. (**e**) Muc2 mRNA expression (n = 5). (**f**) Agr2 mRNA expression (n = 5). (**g**) Muc1 mRNA expression (n = 5). (**h**) Muc3 mRNA expression (n = 5). (**i**) Muc4 mRNA expression (n = 5). (**j**) Muc13 mRNA expression (n = 5). Anterior gradient 2 (Agr2); mucin 2 (Muc2); mucin 1/3/4/13 (Muc1, Muc3, Muc4, and Muc13). * *p* < 0.05; ** *p* < 0.01; *** *p* < 0.001; **** *p* < 0.0001; ns: represents no significance between the values. Control: 0.9% NaCl gavage; LPS: 0.9% NaCl gavage; IgY: unembedded IgY gavage; DE: W/O/W gavage; IgY + DE: W/O/W gavage with embedded IgY.

**Figure 5 foods-13-04138-f005:**
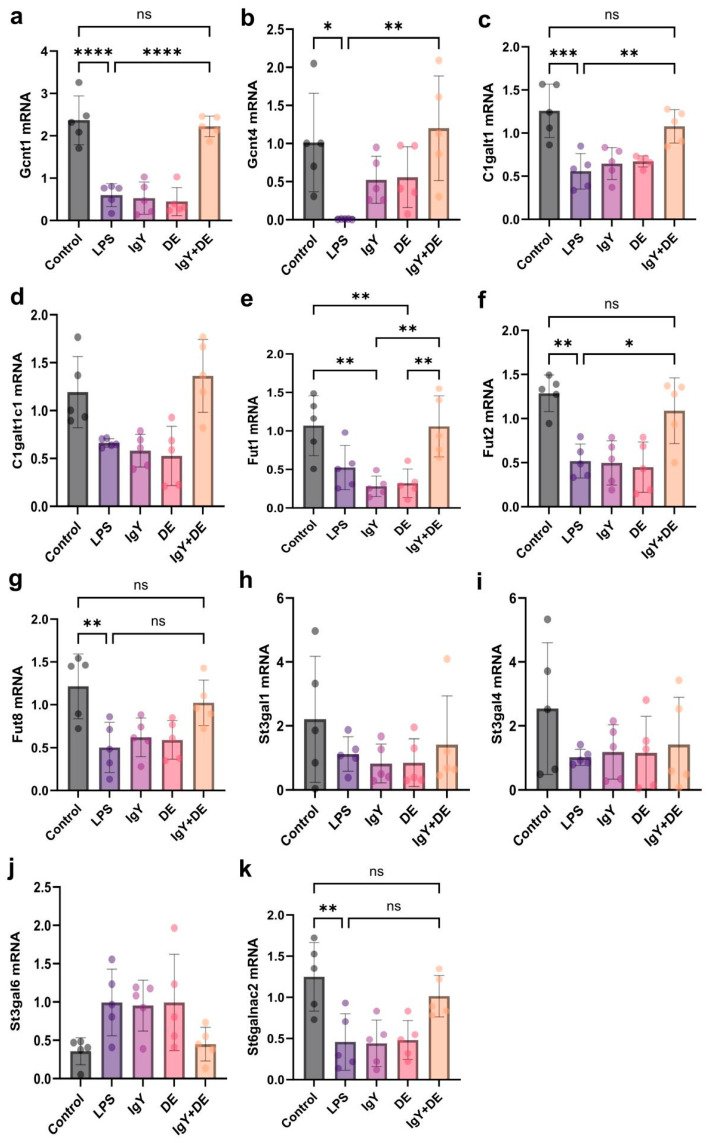
IgY + DE increased the expression of glycosyltransferases involved in mucin glycosylation. (**a**) Gcnt1 mRNA expression (n = 5). (**b**) Gcnt4 mRNA expression (n = 5). (**c**) C1galt1 mRNA expression (n = 5). (**d**) C1galt1c1 mRNA expression (n = 5). (**e**) Fut1 mRNA expression (n = 5). (**f**) Fut2 mRNA expression (n = 5). (**g**) Fut8 mRNA expression (n = 5). (**h**) St3gal1 mRNA expression (n = 5). (**i**) St3gal4 mRNA expression (n = 5). (**j**) St3gal6 mRNA expression (n = 5). (**k**) St6galnac2 mRNA expression (n = 5). (**a**–**d**) glucosaminyl (N-acetyl) transferase 1 (Gcnt1), glucosaminyl (N-acetyl) transferase 4 (Gcnt4), core 1 synthase, glycoproteinN-acetylgalactosamine 3-beta-galactosylt- ansferase 1 (C1galt1), C1GALT1 specific chaperone 1 (C1galt1c1), (**e**–**g**) fucosyltransferase 1/2/8 (Fut1, Fut2, Fut8), (**h**–**k**) ST3 beta-galactoside alpha-2,3-sialyltransferase 1/3/4/6 (St3gal1, St3gal3, St4gal4, St3gal6), ST6 N-acetylgalactosaminide alpha-2,6-sialyltransferase 2 (St6galnac2). * *p* < 0.05; ** *p* < 0.01; *** *p* < 0.001; **** *p* < 0.0001; ns: represents no significance between the values. Control: 0.9% NaCl gavage; LPS: 0.9% NaCl gavage; IgY: unembedded IgY gavage; DE: W/O/W gavage; IgY + DE: W/O/W gavage with embedded IgY.

**Figure 6 foods-13-04138-f006:**
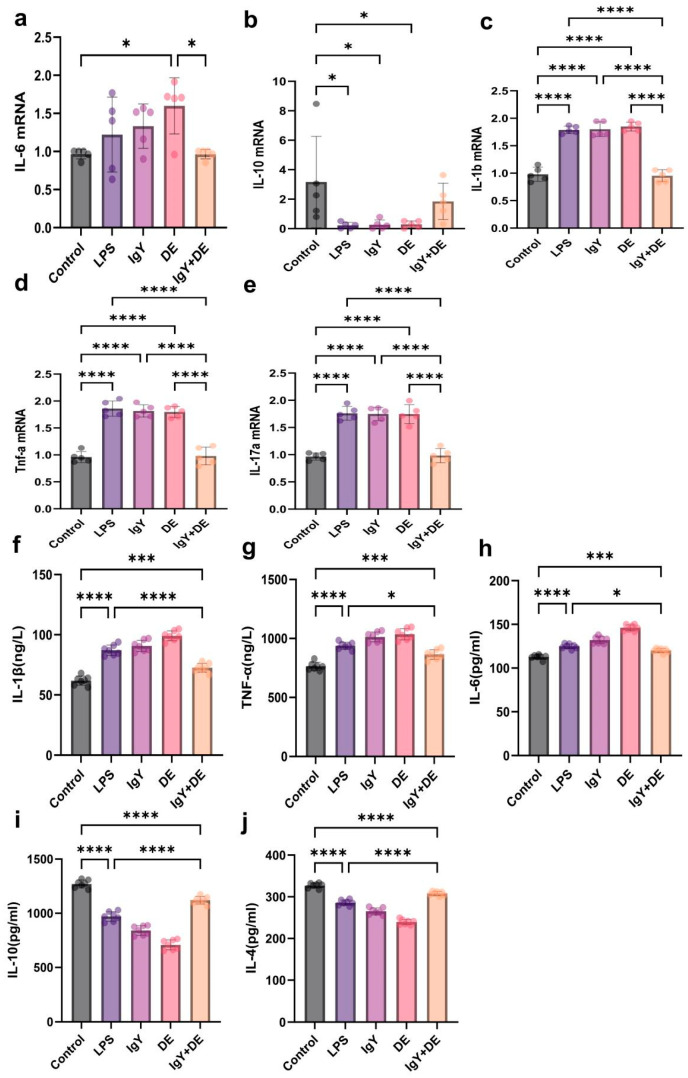
IgY + DE inhibited LPS-induced expression of inflammatory factors in mouse jejunum. (**a**) IL-6 mRNA expression (n = 5). (**b**) IL-10 mRNA expression (n = 5). (**c**) IL-1b mRNA expression (n = 5). (**d**) TNF-a mRNA expression (n = 5). (**e**) IL-17a mRNA expression (n = 5). (**f**) Expression of IL-1β inflammatory factor. (n = 5). (**g**) Expression of TNF-α inflammatory factor. (n = 5). (**h**) Expression of IL-6 inflammatory factor. (n = 5). (**i**) Expression of IL-10 inflammatory factor. (n = 5). (**j**) Expression of IL-4 inflammatory factor. (n = 5) * *p* < 0.05; *** *p* < 0.001; **** *p* < 0.0001. Control: 0.9% NaCl gavage; LPS: 0.9% NaCl gavage; IgY: unembedded IgY gavage; DE: W/O/W gavage; IgY + DE: W/O/W gavage with embedded IgY.

**Table 1 foods-13-04138-t001:** Grouping and dosage of mice administered.

Group Name	Gavage Treatment	Intraperitoneal Injection
Control	0.5 mL 0.9% NaCl	0.2 mL 0.9% NaCl
LPS	0.5 mL 0.9% NaCl	0.2 mL LPS (5 mg/kg)
IgY	0.5 mL IgY (48 mg/mL)	0.2 mL LPS (5 mg/kg)
DE	0.5 mL double-emulsion	0.2 mL LPS (5 mg/kg)
IgY + DE	0.5 mL double-emulsion embedded with IgY (48 mg/mL)	0.2 mL LPS (5 mg/kg)

**Table 2 foods-13-04138-t002:** Primers for real-time PCR.

Primers	Forward	Reverse
*RPL19*	GAAGGTCAAAGGGAATGTGTTCA	CCTGTTGCTCACTTGT
*Math1*	CAAGTGTGTCCAGCAGTGTG	TTGAGTTTCTTCAAGGCGGC
*Spdef*	AGGTGCAATCGATGGTTGTG	AGGGTCTGCTGTGATGTTCA
*EIf3*	CCTATGAGAAGCTGAGCCGA	ACCTCTTCTTCCTTCCAGCC
*KIf4*	GTGCCCCGACTAACCGTTG	GTCGTTGAACTCCTCGGTCT
*Hes1*	CCGGCATTCCAAGCTAGAGA	GGTATTTCCCCAACACGCTC
*Agr2*	GCCAAAGACACCACAGTCAA	CCATCAAGGGTCTGTTGCTT
*Muc1*	GACATCTTTCCAACCCAGGACA	AAGAGAGACTGCTACTGCCATTAC
*Muc2*	ATGCCCACCTCCTCAAAGAC	GTAGTTTCCGTTGGAACAGTGAA
*Muc3*	CCGACACATTGCTGCTGAGAAT	GCTGTCGTCTTGGGTGCTATTT
*Muc4*	CTGTGTCTGAGCTGCCTGTATT	GGGTGTCTGTGTTGATGTTGTTG
*Muc13*	CCCTCATCCTCATCTTGCTGATT	CTCTGCTCTTCTCCATCCTTCTTT
*Gcnt1*	ACAGATTCAGGCTTCCTGTGATT	GCCAGGTGAGATGCCAGTTTA
*Gcnt4*	ATGTCCTGCAGTTCCATTGAGG	ATGTCCTGCAGTTCCATTGAGG
*B 3gnt6*	GGCCAGATTCTCCTCTCTCAAAC	CAGTGTCGTGGGACTCTTGAAC
*C1galt1*	ATGGACACAGTCACCTCAAAGG	GAGGTTCTCAGCAACGTCTATGT
*C1galt1c1*	TCTCACGTCCAAGCCTCGT	TGTGGCCTAGCATAGTGATCAAG
*Fut1*	AGAATTCGCTTGCACCACCA	AAGAAGGAGCCGGCAGAGA
*Fut2*	TGAACTTTCGGCTAAGGTACATCT	GGAAGTGGGCCAGAGGAAAG
*Fut8*	AGGCGAATGGCTGAGTCTCT	TGGCCTTAACAAGCTGTTCTTCT
*St3gal1*	GCCCACTATGCCAGACACTT	TCAGCAGAGTCAAACCCAGC
*St3gal3*	TGCTGCGGTCATGTAGGAAA	CAGCGGAGTCAAGGGAAAGA
*St3gak4*	GGCTCTGGTCCTTGTTGTTG	TCCCTAGAACGGTTGCCAAAA
*St3gal6*	CACCCCAAAAGCGCAGATTTATT	CCTGCCTGAAACAGAGTCCAA
*St6galnac2*	CGGATGTTGTTGCTCGTTGC	AGTCGGCTCTTTCTGTTTTCC
*IL-6*	CACTTCACAAGTCGGAGGCT	CTGCAAGTGCATCATCGTTGT
*IL-1b*	TCTTTGAAGTTGACGGACCC	TGAGTGATACTGCCTGCCTG
*IL-10*	GCTCTTACTGACTGGCATGAG	CGCAGCTCTAGGAGCATGTG
*IL-17a*	TGAGCTTCCCAGATCACAGA	TCCAGAAGGCCCTCAGACTA
*Tnf-a*	GCACAGAAAGCATGATCCGC	CCCCATCTTTTGGGGGAGTG
*Mcp-1/Ccl2*	CAGGTCCCTGTCATGCTTCT	TCTGGACCCATTCCTTCTTG

## Data Availability

The original contributions presented in the study are included in the article, further inquiries can be directed to the corresponding authors.

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
