# Peer review of "Amelioration of LPS-Induced Jejunum Injury and Mucus Barrier Damage in Mice by IgY Embedded in W/O/W Emulsion"

_foods, 2024, doi:10.3390/foods13244138_

Round 1
Reviewer 1 Report
Comments and Suggestions for Authors
Dear Authors,
I read with interest your work, a continuation of your recent article published in the journal Nutrients. The topic is of great interest and opens up perspectives for the possible use of W/O/W-embedded chicken yolk immunoglobulin in treatment of inflammation and jejunal injury.
The article is easy to read despite some errors and inaccuracies that I have pointed out to you. It also needs to be supplemented in its methods and conclusions (see attached file).
Therefore, my opinion is that the paper need improved with the following indications:
REVISIONS
1)Please check and correct the repetitions in Lines 51- 55 of the Introduction.
2)Please make explicit the meaning of W/O/W in the text the first time it is encountered in the reading
3)Please write in Lines 46 in italics the names of the bacteria
4)In Line 5 what is [17]? Is ref [17] incorrect?
5)Please give some background on the AB-PAS technique and give meaning to the acronym
6)Please check and correct the repetition in lines 126-127 of paragraph 2.5.
7)Please introduce a paragraph in the “Materials and Methods” section explaining how the quantification of inflammatory proteins (IL- 1b, TNF-a, IL- 6, IL- 10, IL- 4) was carried out
8)Check all acronyms included in the paper and to express them in the full form when first written in the manuscript (e.g. DE, LPS)
9)Please explain and correct the meaning of the y-axis (PAS*goblet cells/20xfeild) in the caption of Figure 2b
10)Please why is Hes1 not mentioned in the text describing Figure 2 in Section 3.2?
11)Please indicate the meaning of IOD values in the caption of figure 3
12)Please in the sentence in Lines 242-243 delete “that” after show
13)Please review the sentence in lines 267-273 of paragraph 3.5. You have reversed TNF-a with IL-10. Please also add in the test of paragraph 3.5. a sentence differentiating the diversity of experiment between group a-e and group f-j in Figure 5
14)Please enrich your conclusions e.g. by specifying what other steps would be necessary for the therapeutic treatment of jejunum inflammation with W/O/W-embedded IgY
Author Response
1. Please check and correct the repetitions in Lines 51- 55 of the Introduction.
Response: Thank you very much for all of your suggestions, these valuable tips make the article well organized. We've removed the duplicates.
Location: page 2, line number 51-54.
2. Please make explicit the meaning of W/O/W in the text the first time it is encountered in the reading.
Response: Thank you for your suggestion. We added the meaning of W/O/W
Location: page 1, line number 19.
3. Please write in Lines 46 in italics the names of the bacteria.
Response: Thank you for your valuable advice. We changed the name of the bacteria to italics.
Location: page 2, line number 46.
4. In Line 5 what is [17]? Is ref [17] incorrect?
Response: Thank you for your suggestion. ref [17] is correct.
Location: page 2, line number 81.
5. Please give some background on the AB-PAS technique and give meaning to the acronym.
Response: Thank you for your valuable advice. We describe the background of AB-PAS technology and the meaning of the acronym.
Location: page 4, line number 128-133.
6. Please check and correct the repetition in lines 126-127 of paragraph 2.5.
Response: Thank you for your suggestion. We fixed the mistake.
Location: page 4, line number 150.
7. Please introduce a paragraph in the “Materials and Methods” section explaining how the quantification of inflammatory proteins (IL- 1b, TNF-a, IL- 6, IL- 10, IL- 4) was carried out.
Response: Thank you for your suggestion. We added quantification of inflammatory proteins (IL- 1b, TNF-a, IL- 6, IL- 10, IL- 4) to 2.6
Location: page 5, line number 156-160.
8. Check all acronyms included in the paper and to express them in the full form when first written in the manuscript (e.g. DE, LPS)
Response: Thank you for your valuable advice. We increased the full expression of DE, LPS in the paper.
Location: page 1, line number 18-19.
9. Please explain and correct the meaning of the y-axis (PAS*goblet cells/20xfeild) in the caption of Figure 2b
Response: Thank you for your suggestion. We modified Figure 2b and shed light on the meaning of the y-axis
Location: page 8, line number 211-212
10. Please why is Hes1 not mentioned in the text describing Figure 2 in Section 3.2?
Response: Thank you for your valuable advice. We have added a description of Hes1
Location: page 7, line number 203.
11. Please indicate the meaning of IOD values in the caption of figure 3
Response: Thank you for your suggestion. We have added an explanation of IDO
Location: page 10, line number 144-246.
12. Please in the sentence in Lines 242-243 delete “that” after show
Response: Thank you for your valuable advice. We deleted “that”.
Location: page 11, line number 271.
13. Please review the sentence in lines 267-273 of paragraph 3.5. You have reversed TNF-a with IL-10. Please also add in the test of paragraph 3.5. a sentence differentiating the diversity of experiment between group a-e and group f-j in Figure 5
Response: Thank you for your suggestion. We made changes to this paragraph.
Location: page 13, line number 290-298.
14. Please enrich your conclusions e.g. by specifying what other steps would be necessary for the therapeutic treatment of jejunum inflammation with W/O/W-embedded IgY
Response: Thank you for your valuable advice. We have modified our conclusions.
Location: page 17, line number 450-454.
Reviewer 2 Report
Comments and Suggestions for Authors
This innovative approach demonstrates an effective and safe solution to treat intestinal damage caused by lipopolysaccharides (LPS) related to inflammatory diseases such as inflammatory bowel disease and sepsis. IgY as a dietary supplement represents a step towards natural and non-invasive alternatives in disease management. The remarkable thing about the study is the improvement in the bioactivity and stability of IgY thanks to its encapsulation.
I have reviewed the manuscript and found some inconsistencies in the methodology wording. I consider it necessary to fully describe the experiment's setup in terms of the action times of the treatments applied to the mice. I suggest separating the ethical statement. The way it is worded is confusing and omits several conditions. The authors could also include a diagram detailing this part of the methodology.
On the other hand, it is necessary to extend the conclusion by including how these findings could be applied in a broader context, such as in humans or the food industry.
Mention the possible contribution to the development of noninvasive and more accessible treatments. Change phrases such as "These findings demonstrate..." to "This study establishes..." to emphasize the robustness of the results. Include future implications. Include an explicit reference to the need for studies in more advanced models or humans.
Author Response
1. I have reviewed the manuscript and found some inconsistencies in the methodology wording. I consider it necessary to fully describe the experiment's setup in terms of the action times of the treatments applied to the mice. I suggest separating the ethical statement. The way it is worded is confusing and omits several conditions. The authors could also include a diagram detailing this part of the methodology.
Response: Thank you very much for your valuable suggestions, which helped us to organize the article in a clearer and more logical way. We have placed the ethical statements in the manuscript separately. We have added Figure 1 to fully describe the experimental setup and to make the process of drug administration in mice clearer. We have also added Table 1 to illustrate the method of administration and the doses administered.
Location: page 3, line number 97-124.
2. On the other hand, it is necessary to extend the conclusion by including how these findings could be applied in a broader context, such as in humans or the food industry.Mention the possible contribution to the development of noninvasive and more accessible treatments.
Response: Thank you for your valuable advice. We extend the conclusions that W/O/W-embedded IgY is used in clinical applications and in the food industry.
Location: page 17, line number 447-452.
3. Change phrases such as "These findings demonstrate..." to "This study establishes..." to emphasize the robustness of the results. Include future implications. Include an explicit reference to the need for studies in more advanced models or humans.
Response: Thank you for your suggestion. We have changed the phrase ‘These results suggest that ...... ‘phrases such as ‘This study confirms ......’ . The need for research on more advanced models or humans is explicitly mentioned in the conclusions section.
Location: page 17, line number 447-452.
Reviewer 3 Report
Comments and Suggestions for Authors
The reviewed manuscript entitled: “Amelioration of LPS-induced jejunum injury and mucus barrier damage in mice by IgY embedded in W/O/W emulsion”, presents interesting research results, but in my opinion requires some additions, especially in the experimental part, to ensure reliability and repeatability of the presented research. Below are detailed comments on the manuscript.
Abstract: Please do not use abbreviations that are not explained (e.g. LPS, DE)
Materals and methods:
Point 2.1. Line 83: What were the parameters of the process using the high-shear mixer (time, rotations)? Please at least generally present the steps of preparing the emulsion (“A two-step method…”), so that it is not necessary to look for other sources to familiarize oneself with the method.
Point 2.1. What is the composition of the obtained w/o/w emulsion? What component constitutes the oil phase?
Point 2.2. Lines 91 and 92: “All animals were housed under SPF conditions with 12 hours of light and 12 hours of darkness” vs. "The animals were kept in the dark.", so what conditions were the animals kept in?
Point 2.2. What type of probe/gavage was used? In which part of the digestive tract was the sample (emulsion and others) sent? Did it come into contact with hydrochloric acid in the stomach?
Point 2.2. What amounts of fluids (e.g. 0.9% NaCl) were administered? How much emulsion was administered? What dose of IgY was it? Was it a single administration?
Point 2.6. Please specify the conditions for tissue homogenization.
Point 2.7. Do I understand correctly that in all averages n=10?
Discussion:
As I understand it, the form of the w/o/w emulsion as the IgY carrier was of significant importance for the study. Please provide at least the basic characteristics properties of the emulsion obtained and used in the studies, such as particle size distribution, zeta potential, pH, osmotic pressure.
Author Response
1. Abstract: Please do not use abbreviations that are not explained (e.g. LPS, DE)
Response: Thank you very much for these valuable suggestions, which helped us to revise the manuscript to be more complete and clearer. We have added the full names of LPS and DE to the abstract. Lipopolysaccharide (LPS)、Double Emulsion (DE)
Location: page 1, line number 18-20.
2. Point 2.1. What is the composition of the obtained w/o/w emulsion? What component constitutes the oil phase?
Response: Thank you very much for your suggestion. We've added these terms to the 2.1. The emulsifier PCPR (4% w/w) was added to corn oil to form the oil phase (McLean, Shanghai, China), and an amount of 15 g IgY (Unik, Beijing, China), NaCl (0.2% w/w), and 5 g sorbitol (Biotopped, Beijing, China) was added to the internal aqueous phase.
Location: page 2-3, line number 80-95.
3. Point 2.2. Lines 91 and 92: “All animals were housed under SPF conditions with 12 hours of light and 12 hours of darkness” vs. "The animals were kept in the dark.", so what conditions were the animals kept in?
Response: Sorry about the error. The animals kept in SPF conditions with 12 hours of light and 12 hours of darkness.
Location: page 3, line number 102-104.
4. Point 2.2. What type of probe/gavage was used? In which part of the digestive tract was the sample (emulsion and others) sent? Did it come into contact with hydrochloric acid in the stomach?
Response: Thank you for your valuable advice. Mice were gavaged using straight tip 10 gavage pins (GWZ110). The samples were delivered to the stomach of the mice in contact with gastric hydrochloric acid.
Location: page 3, line number 105-111.
5. Point 2.2. What amounts of fluids (e.g. 0.9% NaCl) were administered? How much emulsion was administered? What dose of IgY was it? Was it a single administration?
Response: Thank you very much for your question. We added Figure 1 and Table 1 to represent the experimental procedure. Mice were given the drug by continuous gavage for 14 days. The dose administered per gavage was 0.5 ml. The dose of IgY was 48 mg/mL.
Location: page 3, line number 105-111; page 3, lines 116-124.
6. Point 2.6. Please specify the conditions for tissue homogenization.
Response: Thank you very much for your suggestion. 100 mg of jejunal tissue and PBS used for tissue homogenization. The homogenate was ground at 65 Hz for 5 min.
Location: page 5, line number 154-155.
7. Point 2.7. Do I understand correctly that in all averages n=10?
Response: The n=10 means that a total of 10 mice in a treatment group participated in the gavage and intraperitoneal injection treatments.
8.Discussion:
As I understand it, the form of the w/o/w emulsion as the IgY carrier was of significant importance for the study. Please provide at least the basic characteristics properties of the emulsion obtained and used in the studies, such as particle size distribution, zeta potential, pH, osmotic pressure.
Response: Thank you very much for your valuable suggestions. The basic properties of IgY carrier water-in-oil emulsions have been published and described in detail in our previous articles. In our previous article, we performed storage stability observation, microscopic observation, particle size detection, stability analysis, and in vitro simulated gastrointestinal fluid assay on the constructed double emulsion for encapsulating IgY, and proved that the double emulsion is an effective material for encapsulating IgY. Details are in sections 3.1-3.3 of the previous article. References[16].Wang, Z., et al., Protective effect of IgY embedded in W/O/W emulsion on LPS enteritis-induced colonic injury in mice. Nutrients, 2024. 16(19): p. 3361.
Round 2
Reviewer 3 Report
Comments and Suggestions for Authors
Thank you for answering all my questions and taking my comments into account.